# An Appraisal of the Tissue Injury and Repair (TIAR) Theory on the Pathogenesis of Endometriosis and Adenomyosis

**DOI:** 10.3390/biom13060975

**Published:** 2023-06-11

**Authors:** Marwan Habiba, Giuseppe Benagiano, Sun-Wei Guo

**Affiliations:** 1Department of Health Sciences, University of Leicester and University Hospitals of Leicester, Leicester LE1 5WW, UK; 2Faculty of Medicine and Dentistry, Sapienza, University of Rome, 00161 Rome, Italy; giuseppe.benagiano@uniroma1.it; 3Research Institute, Shanghai Obstetrics & Gynecology Hospital, Fudan University, Shanghai 200011, China; 4Shanghai Key Laboratory of Female Reproductive Endocrine-Related Diseases, Fudan University, Shanghai 200011, China

**Keywords:** adenomyosis, archimetra, archimetrosis, archimetrium, endometriosis, pathogenesis, tissue injury and repair

## Abstract

As understanding their pathogenesis remains elusive, both endometriosis and adenomyosis are often referred to as “enigmatic diseases”. The uncertainty and heightened interest are reflected in the range of expressed views and opinions. There is a sense of urgency because of the entailed patient suffering. The plethora of opinions calls for a critical analysis of proposed theories, both old and new. A series of papers published since 2009 proposed that both endometriosis and adenomyosis originate from the same aberrations occurring within the uterus. This came to be recognized as the tissue injury and repair theory, and the newly coined term “archimetrosis” posits that the two diseases share the same origin. While the theory opens an interesting channel for exploration, its claim as a unifying theory necessitates a critical appraisal. We, thus, undertook this review of the theory and analyzed its underpinnings based on a comprehensive review of the literature. Our appraisal indicates that the theory is open to a range of criticisms. Chief among these is the need for confirmatory evidence of features of abnormal uterine contractility and the lack of data addressing the question of causality. In addition, the theory has, as yet, no supporting epidemiological evidence, which is a major weakness. The theory suffers as it is not open to the test of falsifiability, and it lacks the ability to make useful predictions. It has not addressed the questions, such as why only a small percentage of women develop adenomyosis or endometriosis, given the ubiquity of uterine peristalsis. On the other hand, the triggers and prevention of hyper- or dys-peristalsis become critical to a theory of causation. We conclude that additional supportive evidence is required for the theory to be accepted.

## 1. Introduction

Both endometriosis and adenomyosis are often referred to as “enigmatic diseases” [1,2] since understanding their pathogenesis remains elusive. There is a deluge of reviews on this topic, as evidenced by the 2898 PubMed-indexed papers on “endometriosis and pathogenesis” and the 268 papers on “adenomyosis and pathogenesis”. This reflects an inordinate interest in this topic, considering that the total number of papers on endometriosis or adenomyosis is 34,730 (https://pubmed.ncbi.nlm.nih.gov/?term=endometriosis+or+adenomyosis&sort=date, accessed on 2 April 2023). The outpouring of views and opinions unmistakably signals a heightened interest, as well as an urgency, and necessitates an objective and critical analysis to enable a better understanding of emerging theories, especially when their limitations and possible shortcomings (e.g., lack of proof, missing links, insufficient evidence) are subtle or not explored. Indeed, science thrives in scholarly discourse, debate, and self-correction but stagnates when there is no challenge, dissent, critique, or objective appraisal.

For this reason, we first carried out an in-depth, critical analysis of the theory that endometriosis and adenomyosis develop from abnormally aggressive eutopic endometrium [3]. Here we provide a critical appraisal of the theory known as “Tissue Injury and Repair” (TIAR), also recently integrated as the “Archimetra theory” put forward to explain the pathogenesis of both adenomyosis and endometriosis. It is purported to cover more grounds than Sampson’s retrograde menstruation theory for endometriosis.

Starting in 2009, a series of articles was published proposing that both endometriosis and adenomyosis originate from the same aberration that occurs within the uterus. The hypothesis was developed into a unified theory on the pathogenesis of both conditions and came to be recognized as the TIAR theory [4,5,6]. An invention patent based on the use of CXCL12 (C-X-C Motif Chemokine Ligand 12) and/or CXCR4 (C-X-C motif chemokine receptor 4) levels as markers of TIAR for the early or preliminary stage and/or increased risk of adenomyosis or endometriosis was submitted by Dr. Leyendecker (WO 2019/106034). Although the patent application seems to have been abandoned, the patent formulated around the TIAR theory was purported to identify women with a high risk of developing endometriosis or adenomyosis and raised the prospect of nipping off the bud of the two challenging diseases.

Both endometriosis and adenomyosis are relatively common among women of reproductive age and are leading causes of pelvic pain, infertility, and menstrual disorders. They impact negatively on patients’ quality of life, productivity, and well-being [7,8]. Endometriosis, and perhaps adenomyosis as well, entails a heavy economic burden to society as a whole [9]. Therefore, any proposed hypothesis on pathogenesis might invite interventions aimed at prevention, better management, or cure, and if misdirected, such an effort can result in the wastage of precious resources and/or lost opportunity.

In this connection, it seems important to stress that all existing theories have weaknesses and are open to criticism. Even Sampson’s widely accepted retrograde menstruation theory fails to account for the huge gap between the universality of retrograde menstruation and the prevalence of endometriosis.

The debate about the pathogenesis of endometriosis started with the discovery of the disease. Sampson [10] acknowledged that his theory cannot explain all disease phenotypes. The diverse manifestations led Sampson to classify endometriosis into several phenotypes, with separate pathogenesis: (1) the ectopic endometrial tissue raises by direct extension into the uterine wall (today, this variant is recognized as adenomyosis); (2) peritoneal and ovarian implantation as a result of retrograde menstruation; (3) transplantation of the ectopic tissue, such as through surgical wounds; (4) “metastasis” of endometrial tissue through the vasculature; (5) instances where the presence of ectopic endometrium is developmentally determined [11].

The observation that the high incidence—in fact, a near ubiquity—of retrograde menstruation [12,13] is not matched by the prevalence of endometriosis [14] adds an important factor to the debate about pathogenesis. Against this background, the proposal made by Leyendecker and his associates that endometriosis and adenomyosis are diseases of the “archimetra” (defined as the endometrial-sub-endometrial unit) [15,16] and that they arise through a process of TIAR [4,6], is quite novel. In fact, the new term, archimetrosis, was coined to indicate that the two diseases are one and the same [5], raising the prospect that similar preventive measures might be possible.

It should be noted that the TIAR hypothesis, as proposed by Leyendecker and his associates, should not be confused with ReTIAR or repeated tissue injury and repair hypothesis [17] since the former regards the pathogenesis while the latter focuses on the impetus for the progression of the lesions due to cyclic bleeding in the ectopic endometrium [18].

In this article, we present a critical review of the hypothesis and analyze its underpinnings.

## 2. Materials and Methods

Search Strategy and Inclusion Criteria

This review includes all papers published in PubMed till 1 January 2023. Search items were as follows: (1) uterine contractions, peristalsis, hyper-peristalsis, and dys-peristalsis. This identified 1087 articles that were searched manually to identify any with relevance to endometriosis or adenomyosis: (2) tissue injury and repair (TIAR) and uterus and endometriosis or adenomyosis. This identified 262 articles which were searched manually to identify any papers relevant to the theory of disease causality; (3) uterine development (search for articles published after the review by Habiba et al. [19]). PubMed search engines were used to identify papers meeting the inclusion criteria. Search results were screened based on the title and abstract. Additionally, the reference lists of all papers eligible for review were manually checked for relevant articles.

## 3. Results

### 3.1. Uterine Development and Structure: The Archi- and Neo-Metra Theory

In order to critically appraise the new theory, it is necessary to briefly review our current knowledge of the uterine muscular structure and then analyze the process of its ontology and development.

### 3.2. Structural Aspects

There is evidence that the myometrium is not uniform. The orientation of the muscle fibers in the uterus has been reported to vary from the innermost layers, where the muscle direction is mostly circular, to the interconnected crisscross middle and main layer of the uterus. Leyendecker and co-workers referred to the main component (the middle layer) layer as the *stratum vasculare* [20]. This separates the subserosa (the outermost layer) or the *stratum supravasculare* from the innermost layer, which they referred to as the *stratum subvasculare* (Figure 1A). However, the boundaries between these layers are not defined histologically [20]. Structural studies have shown that the transition in the myometrial layers is gradual, with no discernable histological demarcation between the inner and outer zones [21]. As an example, the concept of the junctional zone that designates the endometrium–myometrium interface (EMI) is based on MR imaging [22], not on histologically identifiable features. The classic studies of uterine vasculature, such as those reported by Sampson, show that the rich vascular network of anastomosis that contains the arcuate arteries lies between the outer and middle third of the myometrium [23,24,25]. Blood vessels run medially from that network to supply the myometrium and terminate by supplying the endometrium. Based on its vasculature, Sampson divided the myometrium into three zones: the peripheral or outer third, which is supplied by the peripheral arteries; the arcuate zone, which is the narrow area containing the main vessels; the radial zone, which corresponds to the inner two-thirds of the myometrium, supplied by the radial vessels. This vascular distribution is at variance with the description of the stratum vasculare as forming the main bulk of the myometrium (Figure 1).

Leyendecker et al. [26] proposed to designate as “archimetra” the endometrial–subendometrial region together with the main bulk of the muscle layer (the stratum subvasculare) and the term “neometra” to be applied to the outer layers of the myometrium. They argued that only the archimetra is of paramesonephric origin and that the neometra is of non-Müllerian origin. However, a different embryonic origin is difficult to substantiate. It is commonly accepted that the mesonephric (Wolffian) duct first develops from the intermediate mesoderm and is critical to the development of the paramesonephric (Müllerian) duct [27,28]. Epithelial cells of the Müllerian duct develop adjacent to the rostral mesonephric epithelium as invaginations of the coelomic epithelium [29]. The mesenchyme that surrounds the Müllerian duct epithelium is derived from mesonephric (Wolffian) mesenchyme and coelomic epithelial cells localized along the length of the mesonephros [30,31]. Thus, the myometrial layers in the context of “archimetra” and “neometra”, as depicted by Leyendecker et al. [20], are at direct odds with that depicted by Sampson [24].

There is also evidence that the inner myometrium can develop from endometrial stroma and of the potential existence of smooth muscle metaplasia, i.e., endometrial stromal cells can be coaxed to transdifferentiate to smooth muscle cells [32,33,34]. These findings suggest that the whole myometrium shares a common embryonic origin. It is also well-recognized that adenomyosis is not confined to the inner or to the mid-myometrial layers. Both observations challenge the view that endometriosis and adenomyosis are diseases of “archimetra”.

The archimetra theory is partly based on the work of Werth and Grusdew [35], who reported on the features of uterine development from the fetal stage to maturity. This study included five samples from the end of the third month to the fifth month of gestation. This publication, however, does not contain any claims as to the mesonephric/paramesonephric origin of the myometrium. Interestingly, the authors observed that comparative studies with other species (which they referred to as “genetic studies”, employing the terminology used at the time) add little, if any, to our understanding of the mature human uterus [35].

Two additional publications adopted the notion that adenomyosis and endometriosis are “diseases of archimetra” [36,37]. However, neither of these studies added any original information to support the archimetra hypothesis beyond the TIAR theory.

It has been recently suggested that the stratum vasculare in humans constitutes the main musculature of the uterus and that its (hyper)-contraction manifests as primary dysmenorrhea and leads to uterine injury and also the expulsion of the basal endometrium and the development of endometriosis [5]. In support of this hypothesis, it was proposed that the human uterine structure is unique [5]. However, there are considerable similarities between the structure of the myometrium in the mouse and the human [38]. While it is plausible that the mesh-like structure of the middle layer of the human myometrium evolved secondarily to the fusion between the Müllerian ducts, recent research using three-dimensional reconstruction of the mouse uterus identified a middle myometrial layer that connects the inner circular and the outer longitudinal muscle layers in the bicornuate uterus [39]. This finding is critical as it demonstrates that the three-layer myometrium is not unique to Haplorrhines (one of the two suborders of primates, which includes monkeys, apes, and humans) but that it is also shared with some Euarchontoglires (a superorder of mammals, also called “supraprimates”, that includes rodents. See Figure 2). This raises doubt about the proposed notion that the human myometrial structure played a unique role in primate evolution [5]. 

On the other hand, it is not clear whether there is a causal link between myometrial structure and menstruation, and there is no direct information available to compare the myometrium in the majority of mice species, which do not menstruate and the Egyptian spiny mouse (*Acomys cahirinus*)*,* the only mouse species known to menstruate spontaneously. Thus, the claim that the human uterine structure is unique needs to be substantiated, particularly given that adenomyosis does occur *spontaneously* in the animal kingdom [40]. Furthermore, endometriosis also occurs spontaneously in primates, such as baboons [41] and cynomolgus monkeys [42], as well as in mice with oncogenic mutations [43]. In other words, the putative uniqueness cannot be used as a justification for the hypothesis.

### 3.3. The Archimetra Theory

An important facet of the “archimetra theory” is that adenomyosis and endometriosis are two phenotypes of the same disease. The idea is certainly not new, and, at one time, it was proposed by our group [44,45] based on the observation that the two diseases are characterized by dysfunction in both the eutopic and heterotopic endometrium. A recent review [46] collated the similarities, as well as differences, between the endometrium in adenomyosis and endometriosis. It is also recognized that the two conditions often coexist.

In this regard, it should be borne in mind that, even if proven, a common origin does not equate with endometriosis and adenomyosis, representing the same disease. The commonality of structure also calls into question the value of the term archimetrosis. Adenomyosis and endometriosis differ in their gross, histological, clinical manifestations, and risk factors, including patient age, parity, and the history of iatrogenic uterine procedures in adenomyosis [47,48,49,50,51,52]. This suggests that, at the very least, the triggering events for the two diseases are likely to be different.

### 3.4. The Tissue Injury and Repair Theory

An essential component of the unifying theory of adenomyosis and endometriosis [4,6] involves differences in local production of estrogens in the eutopic and ectopic endometrium in affected women.

The theory seems to be based largely on the premise that endometriosis and adenomyosis are caused by trauma due to chronic uterine peristaltic activity (or to phases of hyperperistalsis) and that this induces micro-traumatization at the EMI activating TIAR, resulting in an increased local production of estrogen, which propagates the cycle causing disease. The theory also attempts to account for the absence of endometriosis against the universal phenomenon of retrograde menstruation. One proposition is that endometriosis occurs in women with hyperperistalsis, which leads to the dislocation of basal endometrium and propagation of a cycle of TIAR [15].

The ability to heal after the injury is fundamental to the survival of all organisms and involves an evolutionarily conserved mechanism of tissue regeneration and repair. Regeneration entails the replacement of damaged tissue through the proliferation of surrounding undamaged tissue. Repair entails the formation of granulation tissue and its maturation in the form of scarring. The endometrium has a unique ability to regenerate after menstrual desquamation without progressive scarring, but this does not seem to be shared by the myometrium, which heals by fibrosis [53,54].

Over the years, the mechanism of TIAR has been gradually advanced (see [4,5,6]) to account for both endometriosis and adenomyosis. The theory is built on the occurrence of “auto traumatization” secondary to uterine hyperperistalsis and that this is self-perpetuating secondary to the release of local estrogen and relevant cytokines. The trigger of hyperperistalsis is said to reside in an increased or prolonged estradiol stimulation leading to prolonged supraphysiological mechanical strain on the cells near the fundo-cornual raphe. The theory postulates that such increased estrogen production is caused by prolonged follicular phase, anovulatory cycles, follicular persistency, or the presence of large antral follicles. While there is a plausible relation between endometriosis and adenomyosis and systemic or local estrogen production, surprisingly, there is no documented evidence for the occurrence of uterine (endometrial or myometrial) trauma or for the complete detachment of the endometrium. An important missing link is the demonstration of the time sequence of TIAR; in other words, it has not been shown that the presence of different stages of TIAR, including hemostasis, inflammation, proliferation, and remodeling, can precede the implantation of ectopic endometrium, or exist in its absence. The extent of endometrial shedding at menstruation may vary, and there is no convincing evidence that this varies in women with or without endometriosis [55,56,57].

At the core of the TIAR theory is a local, injury-induced overproduction of estrogen [4,5,6]. Unfortunately, most of the arguments presented to support the theory [4,5,6] are based on the feed-forward loop model that is true for *ectopic* endometrium [58], not myometrium or normal endometrium, as elaborated previously [59]. Of course, injury involving the vasculature may lead to estrogen production due to platelet activation [60,61], but whether hyperperistalsis can lead to such injury is unclear. Overall, there is no supportive experimental evidence.

### 3.5. The Possible Role of Hyperperistalsis

Our literature search identified numerous articles that addressed uterine peristalsis and contractility in relation to sperm transport and embryo implantation, but only eight addressed the possible relation between peristalsis and endometriosis or adenomyosis (Table 1). Uterine peristalsis is a universal phenomenon in women of reproductive age and, as such, cannot account for the occurrence of disease in some, but not all, women.

The Leyendecker’s group [62,63,64,65,66,67,68,69,70,71] investigated uterine peristalsis in various situations and reported that it was increased in women with endometriosis compared to controls. The frequency of contractions almost doubled in the early, mid-, and late-follicular and in mid-luteal phases of the cycle. However, there was no statistically significant difference in the frequency of contraction in the late luteal or in the menstrual phases of the cycle. Despite the absence of a demonstrable increase in the frequency or magnitude of contractions at the relevant phase of the cycle, the authors concluded that uterine hyperperistalsis preceded retrograde menstruation and constituted the mechanical cause of endometriosis.

In summary, before the hypothesis of a major role of hyperperistalsis in the pathogenesis of endometriosis can be considered, it is necessary to demonstrate *all* the following issues:(1)Evidence of hyperperistalsis in the majority of women with endometriosis;(2)Such hyperperistalsis *precedes* the occurrence of the disease;(3)Endometriosis occurs mostly in women with hyperperistalsis;(4)Women with hyperperistalsis have a significantly higher risk of developing endometriosis.

In addition, while retrograde menstruation has been proposed as a mechanism for endometriosis, it would be important to define the role of “hyper”-peristalsis and show that it is a requirement above that of “normal” peristalsis leading to the expulsion of viable basal endometrial fragments into the pelvic cavity via retrograde menstruation. As only circumstantial evidence exists on the first point, doubt must remain as to the relevance of the phenomenon to endometriosis.

The weakness of the relationship between the establishment of endometriotic lesions and increased peristalsis is reflected in articles of Leyendecker et al. [5,6], who wrote that given the high prevalence of endometriosis and adenomyosis and the apparent lack of a “causal event” leading to hyperperistalsis, “it appears to be unavoidable that, with time, chronic normoperistalsis throughout the reproductive period of life leads to the same extent of micro-traumatization”. However, Kunz et al. [71] and Leyendecker et al. [63] proposed that persistent infertility after medical or surgical treatment of endometriosis is due to impairment of sperm transport through abnormal peristalsis. In other words, the problem resides in the misdirection, not the magnitude of the uterine peristalsis.

The evidence for the occurrence of “hypercontractility” or “hyperperistalsis” is derived from the study of women with or without endometriosis, using vaginal sonographic assessment of contractions and hystero-salpingo-scintigraphy (HSSG) [72]. The study did not comment on the presence or absence of adenomyosis or how that might impact sonographic assessment. The information on uterine contraction as assessed by ultrasound is interesting but awaits further confirmation. HSSG is currently rarely used [73], and research on the rat demonstrates that only viable spermatozoa can reach the ampullary part of the tube [74]. This suggests the need for further understanding of uterine peristalsis and its aberrations and whether these may be causally related to endometriosis.

Notwithstanding this, the demonstration of differences in uterine contractility between women with or without endometriosis does not constitute evidence of a causative link since the aberrations reported could well be the consequence, as opposed to the cause, of endometriosis or adenomyosis.

Last but not least, the exact definition and cut-off that warrants the term “hyperperistalsis” is unclear: it was argued that contraction waves in the uterus in women with endometriosis might start in the middle of the uterine cavity or at different sites at the same time or vanish before reaching the fundus [75]. This renders the transport function ineffective or misdirected. Since it has been consistently demonstrated that endometriosis induced in several animal models led to subsequent endometrial aberrations [76,77,78,79], it is unclear whether endometriosis and, perhaps, adenomyosis as well, can be the cause rather than the result of uterine hyper- or dys-peristalsis.

### 3.6. The Role of Dysperistalsis

Intuitively, *dysperistalsis* could be defined as having uterine contractions that are either asynchronized, irregular, or spasm-like, possibly along with increased frequency and/or amplitude. Indeed, increased myometrial expression of the oxytocin receptor, along with increased uterine contractility, which correlated positively with the severity of dysmenorrhea, was found in women with adenomyosis [80]. In mice with induced adenomyosis, uterine contractions were found to have increased frequency and amplitude, as well as irregularity [81]. This suggests that uterine hyper- and dys-peristalsis can occur as a *result*, as opposed to a *cause*, of adenomyosis. In particular, this uterine hyperactivity can be rectified by drug treatment in mice [81,82]. The ultrastructural analysis also reveals that smooth muscle cells from uteri with adenomyosis are different from smooth muscle cells of normal uteri [83].

According to the definition of Leyendecker et al. [62], “[d]ysperistalsis is defined as contractions originating in the middle portion of the uterus and spreading simultaneously to the fundus and the cervix, or contractions starting simultaneously at different sites, creating a convulsive appearance in uterine activity, with some waves vanishing before reaching the uterine fundus…”. They also reported that the nature of contractions differed in those with endometriosis compared to controls. In endometriosis, contractions appeared to fit the definition of dysperistalsis and arrhythmia. Interestingly, uterine contractions were not reduced at the time of peak progesterone level in the mid-luteal phase compared to the early or mid-follicular phase. This is at variance with a more recent study of uterine peristalsis, which employed 4D ultrasound technology and showed reduced uterine contractility at the time of implantation, perhaps aimed at aiding implantation through quiescence [84].

In addition, the study by Moliner et al. [84] showed no difference between subjects considered with hypercontractility [>1.51 contractions per minute (cpm)] and those with normal contractility (≤1.51 cpm) in terms of patient’s age and the presence of adenomyosis, fibroids, adhesions, or polyps. However, there was an inverse association between contractility and progesterone levels due possibly to the suppression of myometrial NF-κB and COX-2 by progesterone receptors [85]. It is also important to note that the technique of assessing uterine contractility using ultrasound is not fully validated. Moliner et al. [84] reported moderate inter-observer variability when using 4D ultrasound, but no assessment was made by Leyendecker et al. [62]. To date, no other publications have independently confirmed the presence of dysperistalsis in endometriosis. There is no report on the sensitivity analysis or how inter-observer variability might impact the conclusions.

Dysperistalsis, which was argued to occur during the late follicular phase [59], is not temporally related to menstruation. This casts doubts on its relevance to basal endometrial breakdown (which is referred to as the syndrome of dislocated basal endometrium) and its subsequent implantation [62].

It is notable that studies that compared uterine contractions in the presence and absence of endometriosis have reached contradictory conclusions. Bulletti et al. [86] reported the occurrence of uterine contractions during the menstrual phase in the presence or absence of endometriosis, but fewer patients with endometriosis had uterine contractions in the study by Kido et al. [65]. Studies that reported contraction frequency also reached contradictory conclusions [62,65,86,87,88].

In a recent publication, Leyendecker et al. [66] reported a prevalence of endometriosis in 80.6% of cases of adenomyosis and a prevalence of adenomyosis in 91.1% of cases of endometriosis. They argued in favor of extending their concept to include “archimetral compression by neometral contractions” as the predominant cause of uterine auto-traumatization. This hypothesis postulates that the contraction of the neometra leads to rupture of the archi-myometrium at the cornual angles and that fragments of the basal endometrium are dislocated into the myometrial wall. The co-existence of endometriosis and adenomyosis has been consistently reported in the literature (see, for example, [89]). However, over 80% coincidence between adenomyosis and endometriosis in their cohort is surprisingly high and has not been observed elsewhere. Furthermore, a hypothesis that links adenomyosis to compression by the neometra cannot be established based on a link between adenomyosis, endometriosis, and dysmenorrhea. The notion is advanced that excessive uterine contractions lead to dissociation of the basal endometrium leading to the development of endometriosis [15]. Nevertheless, the evidence produced to support this hypothesis is, at best, tenuous [90].

Moreover, the hypothesis does not explain the vast variation in the case of coexisting endometriosis and adenomyosis, nor can it explain why different subtypes of adenomyosis have different links with endometriosis. For example, external, but not internal, adenomyosis is often linked with deep endometriosis [91,92,93].

In conclusion, the proposition that hyper- or dys-peristalsis may cause adenomyosis/endometriosis can be challenged since all data showing hyper- or dys-peristalsis are based on women who have already been diagnosed with endometriosis and/or adenomyosis. We have shown that induced adenomyosis in rodents can cause hyper- and dys-peristalsis [81,82]. As of now, there are no data to show that hyper- or dys-peristalsis indeed *precedes* adenomyosis.

### 3.7. The Lack of Support from Epidemiological Data

When it comes to the search for the pathogenesis/etiology of a disease of interest, epidemiological data often provide the first and most invaluable clue, as seen in AIDS, type 2 diabetes, endometrial cancer, and diethylstilbestrol-induced vaginal cancer, to name a few. For endometriosis, in particular, retrograde menstruation has been shown to be nearly universal [13,94]. For adenomyosis, iatrogenic uterine procedures, such as dilatation and curettage, and induced abortion, have been amply shown to increase the risk of developing the disease later in life [47,48,49,50,51,52]. Yet surprisingly, there have been no epidemiological data whatsoever that suggested that uterine hyper- or dys-peristalsis was the cause of adenomyosis or endometriosis. Conceivably, an episode of TIAR severe enough to initiate the genesis of adenomyosis would at least produce some signs in some women. It would be surprising to see that such occurrences have gone undetected for so long.

## 4. Discussion

### Issues of Falsifiability and the Ability to Make Useful Predictions

Scholarly theories are typically based on a systematic and coherent organization of facts and ideas about various phenomena in a field of inquiry and are meant to understand, explain, and, most importantly, predict. As alluded to in previous research [40], a good theory has to satisfy at least three basic requirements for being falsifiable and explanatory and having the ability to yield useful predictions. It is unclear how the TIAR theory can be tested, let alone be predictive of the diseases.

In the case of the TIAR hypothesis, the following questions need to be addressed: (1) Why does only a small fraction of women develop adenomyosis and/or endometriosis, given that uterine peristalsis is ubiquitous? (2) What is the “*primum movens*” for the hyper- or dys-peristalsis? Can it be prevented or mitigated? (3) Assuming that TIAR is the cause of the two diseases, what are the real triggers of TIAR itself, and what could be done for prevention?

All theories have constructs. In the TIAR theory, the archimetra can be considered an important construct. However, whether neometra and archimetra genuinely exist is still unclear. Another construct may be hyper- or dys-peristalsis, but whether they precede the occurrence of endometriosis or adenomyosis remains to be demonstrated.

The systemic levels of SDF-1 and CXCR4 have been proposed (and were the subject of a now-abandoned patent application) as markers for uterine TIAR. However, elevated serum levels of SDF-1 and CXCR4 are *not* specific to the uterine TIAR. In fact, in many organs, repair of tissue injury that involves the recruitment of stem cells is associated with increased local SDF-1/CXCR4 levels [95].

## 5. Conclusions

While the TIAR theory provided a platform for advancing our knowledge, a critical analysis suggests the need for substantial additional supportive evidence, and the theory remains open to a number of questions. Chief among them is the need for confirmatory evidence of features of uterine contractility and the absence of data addressing the question of causality. In addition, the theory has, as yet no supporting epidemiological evidence. Aside from the issues of falsifiability and the ability to make useful predictions, it has not addressed the question of why only a small percentage of women develop adenomyosis or endometriosis, given the ubiquity of uterine peristalsis. If hyper- or dys-peristalsis are the causes, this raises questions about its trigger and prevention. Thus, significant additional evidence is required before this theory can be considered a credible and viable proposition for the pathogenesis of endometriosis and/or adenomyosis.

To paraphrase Karl Marx, different theories have only explained why or how adenomyosis/endometriosis occurred; the point, however, is to prevent these conditions. The appraisal presented here suggests that this goal still remains elusive. As such, the TIAR theory requires further review, and considering it as a guide to future therapeutics is premature.

## Figures and Tables

**Figure 1 biomolecules-13-00975-f001:**
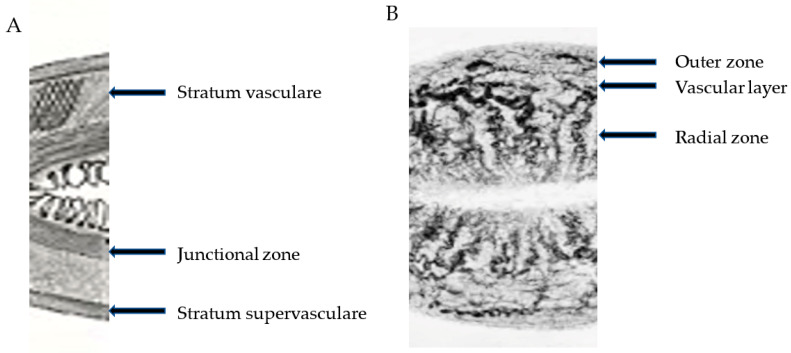
The myometrial layers as depicted by Noe et al. [20] and Leyendecker (WO 2019/106034) (Panel (**A**)) and by Sampson [24] (Panel (**B**)). The study by Sampson was based on histological and vascular casts of the uterine blood vessels. The arcuate arteries pass between the outer third and the inner two-thirds of the uterine wall, which they divide into the peripheral zone nourished by peripheral branches of the arcuate arteries and the inner or radial zone nourished by radial branches that terminate into capillaries in the endometrium. According to Noe and Leyendecker, the stratum vasculare and supravasculare form the neometra. Adapted from Noe et al. [20] and Sampson [24].

**Figure 2 biomolecules-13-00975-f002:**
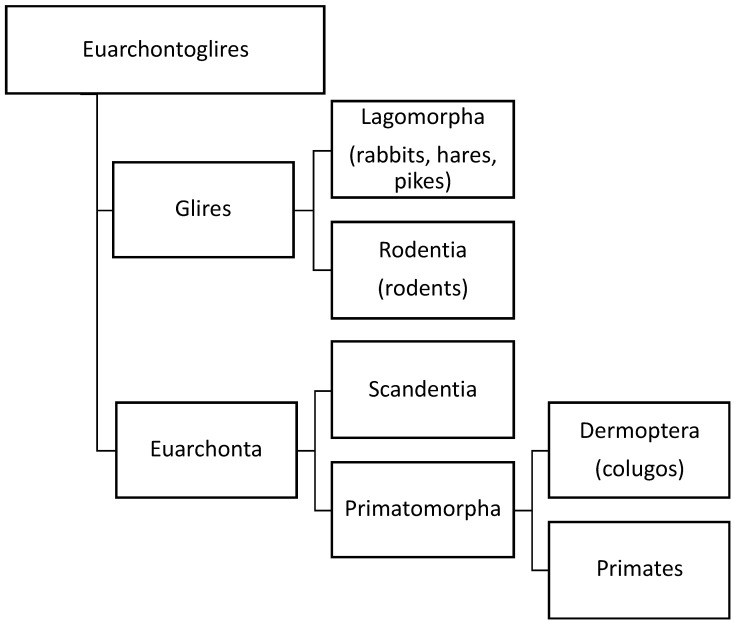
Orders of different species based on our current knowledge of evolution of species. Three-layer myometrium is not unique to Haplorrhines (one of the two suborders of primates, which includes monkeys, apes, and humans), but it is also shared with other Euarchontoglires (a superorder of mammals, also called “supraprimates”, that includes rodents. Adapted from Wikipedia (https://en.wikipedia.org/wiki/Euarchontoglires, accessed on 10 January 2023).

**Table 1 biomolecules-13-00975-t001:** Summary of published findings on uterine peristalsis.

Sequence No.	Reference	Main Findings
1	Leyendecker et al. [62]	Women with endometriosis displayed a marked uterine hyperperistalsis that differed significantly from the peristalsis of the controls during the early and mid-follicular and mid-luteal phases.
2	Kataoka et al. [63]	Thickness of the inner low-signal-intensity myometrial layer and endometrial distortion were significantly associated with pain degree (*p* < 0.001), while uterine peristalsis was undetectable when pain was severe or moderate. The area of the uterine myometrium significantly decreased during cycle days 1–3 in the dysmenorrheic group, as compared with that in the eumenorrheic group (*p* = 0.010). No specific reference to endometriosis or adenomyosis.
3	Kissler et al. [64]	Sperm transport is disturbed by hyperperistalsis when at least one focus of adenomyosis can be detected via magnetic resonance imaging (MRI) and turns into dysperistalsis (a complete failure in sperm transport capacity) when diffuse adenomyosis affecting all myometrial uterine muscle layers is detected.
4	Kido et al. [65]	Uterine peristalsis was identifiable in 3/10, 3/13, and 3/3 of the endometriosis patients in each menstrual cycle, respectively, and in 11/12, 3/12, and 5/12 of their control subjects. Peristaltic detection rate and frequency were significantly lower for the endometriosis group than for the controls in the periovulatory phase only (*p* < 0.05). Sustained contractions were recognized in 19/36 control subjects and in 13/26 endometriosis patients, but the difference was not significant. Uterine peristalsis appears to be suppressed during the periovulatory phase in patients with endometriosis, which may have an adverse effect on sperm transport.
5	Leyendecker et al. [66]	Menstrual “archimetral compression by neometral contraction” has to be considered as an important cause of uterine auto-traumatization in addition to uterine peristalsis and hyperperistalsis.
6	Puente et al. [67]	Transient non-cyclical activity of external myometrium should be considered to avoid errors in diagnosis of adenomyosis and uterine anomalies.
7	Rees et al. [68]	The image quality of the HASTE (half-Fourier acquisition single-shot turbo spin-echo) technique was superior to TVUS, and FISP (fast imaging with steady-state precession) technique when studying uterine peristalsis.
8	Shaked et al. [69]	A software modeling study

## Data Availability

Not applicable.

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
