# Peer review of "An Appraisal of the Tissue Injury and Repair (TIAR) Theory on the Pathogenesis of Endometriosis and Adenomyosis"

_biomolecules, 2023, doi:10.3390/biom13060975_

Round 1

Reviewer 1 Report

Introduction, lines 100-103: Conclusions should be moved to the Conclusions section.

The results of the author’s search and their interpretation are located in the section Materials and Methods. The authors should set apart the section Results and Discussion.  

Author Response

Introduction, lines 100-103: Conclusions should be moved to the Conclusions section.

Reply: Thank you for this suggestion. The paragraph in question has been moved to the conclusion.

The results of the author’s search and their interpretation are located in the section Materials and Methods. The authors should set apart the section Results and Discussion.

Reply: Again, thanks for the suggestion. We have revised the sections as advised.

Reviewer 2 Report

Nice summary of the literature on the Tissue Injury and Repair (TIAR) Theory on the pathogenesis of endometriosis and adenomyosis.

Author Response

Nice summary of the literature on the Tissue Injury and Repair (TIAR) Theory on the pathogenesis of endometriosis and adenomyosis.

Reply: Many thanks, we appreciate the comment and the opportunity to present this work.

Reviewer 3 Report

The authors describe TIAR and endometriosis, and adenomyosis. However, the relationship between endometrial contractions and TIAR is not well understood. It is difficult to understand how uterine contractions can cause tissue damage outside of menstruation. A more detailed explanation of the relationship between contractions, TIAR, and uterine archimetra is needed.

Author Response

The authors describe TIAR and endometriosis, and adenomyosis. However, the relationship between endometrial contractions and TIAR is not well understood. It is difficult to understand how uterine contractions can cause tissue damage outside of menstruation. A more detailed explanation of the relationship between contractions, TIAR, and uterine archimetra is needed.

Reply: Many thanks for this comment. We can only agree with the Reviewer that “A more detailed explanation of the relationship between contractions, TIAR, and uterine archimetra is needed”. As we understand it, the reviewer agrees with the arguments presented in our paper which sought to clarify the nature of the presumed relation between uterine contractions and tissue damage leading to endometriosis and adenomyosis. We conclude that available literature does not provide enough information to clarify this issue and that more evidence is needed for this theory to be accepted.

Reviewer 4 Report

The review “An appraisal of the Tissue Injury and Repair (TIAR) theory on

the pathogenesis of endometriosis and adenomyosisis an interesting analysis

that focuses on the potential role of damage theory in the pathogenesis of adenomyosis and endometriosis. This work is well structured, however, I have a one comment:

On lines 124-125 it is mentioned that the main component of the myometrium is called the "stratum vasculare". At the same time, on lines 157 - 159, the main component is “stratum subvasculare”. In addition, figure 1A does not give a clear answer to the organization of the myometrium. I would like to ask you to clarify the structure of the myometrium.

Author Response

The review “An appraisal of the Tissue Injury and Repair (TIAR) theory on the pathogenesis of endometriosis and adenomyosis” is an interesting analysis that focuses on the potential role of damage theory in the pathogenesis of adenomyosis and endometriosis. This work is well structured; however, I have a one comment:

On lines 124-125 it is mentioned that the main component of the myometrium is called the "stratum vasculare". At the same time, on lines 157 - 159, the main component is “stratum subvasculare”. In addition, figure 1A does not give a clear answer to the organization of the myometrium. I would like to ask you to clarify the structure of the myometrium.

Reply: Many thanks for this useful comment. We acknowledge that the text in its present form can be confusing and have therefore clarified the classification in the text by referring to the location of each of the components.